# In-hospital survival of adults with HIV-associated cryptococcal meningitis in Tanzania: A retrospective comparison of amphotericin B-based regimen and fluconazole monotherapy

Mlela Msongela[1,2], George Musiba[2,3], Juma A. Mohamed[4], Alphonce I. Marealle[2], Manase Kilonzi [2*], Ritah F. Mutagonda [2]

1 Department of Pharmacy, Singida Regional Referral Hospital, Singida, Tanzania, 2 Department of Clinical Pharmacy and Pharmacology, School of Pharmacy, Muhimbili University of Health and Allied Sciences, Dar es Salaam, Tanzania, 3 Department of Pharmacy, Muhimbili National Hospital, Dar es Salaam, Tanzania, 4 Department of Pharmaceutics and Pharmacy Practice, School of Pharmacy, Muhimbili University of Health and Allied Sciences, Dar es Salaam, Tanzania

* manasekilonzi@gmail.com

## Abstract

### Objective

This study aimed to examine treatment modalities, outcomes, and factors associated with in-hospital survival among people with HIV (PHIV) diagnosed with cryptococcal meningitis (CM) in Tanzania.

### Methods

This hospital-based cross-sectional study retrospectively reviewed records of PHIV admitted to medical wards at Dodoma and Singida Regional Referral Hospitals in Tanzania from July 2019 to June 2024. Data on socio-demographics, antiretroviral therapy (ART) status, CD4 count, treatment, and outcomes were extracted using a standardised data collection tool. The primary outcome was in-hospital survival (discharged alive vs died). Descriptive statistics summarised patient characteristics, and modified Poisson regression with robust variance estimated adjusted risk ratios (aRR) for factors associated with being discharged alive.

### Results

A total of 159 PHIV with CM records were reviewed. Of these, 89 (56.0%) were aged 36–55 years, 89 (56.0%) were female, and 138 (86.8%) were in WHO clinical stage IV. Of 159 patients, 88 (55.3%) received fluconazole monotherapy. In-hospital mortality among CM patients was 65 (46.5%). Discharge alive occurred in 61/73 (83.6%) of those on amphotericin B–based regimens versus 13/66 (19.7%) on fluconazole monotherapy. Patients treated with amphotericin B–based regimens were four times

**Data availability statement:** All relevant data are within the paper and its Supporting information files.

**Funding:** The author(s) received no specific funding for this work.

**Competing interests:** The authors have declared that no competing interests exist.

**Abbreviations:** AIDS, acquired immunodeficiency syndrome; aPR, adjusted prevalence ratio; aRR, adjusted risk ratio; ART, antiretroviral therapy; CI, confidence interval; CM, cryptococcal meningitis; CrAg, cryptococcal antigen; cPR, crude prevalence ratio; CSF, cerebrospinal fluid; CTC, care and treatment centre; DRRH, Dodoma Regional Referral Hospital; HCPs, health care providers; HIV, human immunodeficiency virus; LTFU, lost to follow-up; MUHAS, Muhimbili University of Health and Allied Sciences; PHIV, people I with HIV; RRHs, regional referral hospitals; SSA, sub-Saharan Africa; SPSS, Statistical Package for the Social Sciences; SRRH, Singida Regional Referral Hospital; WHO, World Health Organization.

more likely to be discharged alive compared to those on fluconazole monotherapy (aPR = 4.19, 95% CI: 2.46–7.16, p < 0.001).

## Conclusion

CM remains a leading opportunistic infection causing high mortality among PHIV, with most patients managed using fluconazole monotherapy. In-hospital survival was significantly higher with amphotericin B–based regimens, highlighting the need to align practice with guideline recommendations. Further qualitative research is warranted to explore barriers to implementing recommended CM treatment.

## I. Introduction

Cryptococcus meningitis (CM) is the most common opportunistic infection and remains the leading cause of mortality in people with HIV (PHIV) [1]. Globally, CM is estimated to cause 350,000 to 1.5 million new cases annually, causing a mortality of approximately 181,000, with around 75% of deaths being reported in sub-Saharan Africa (SSA). In Africa prevalence of CM in PHIV ranges from 5.1% to 33%, with the majority of these cases and mortality occurring in SSA. For instance, studies conducted in South Africa, Ethiopia, Uganda, and Kenya reported a prevalence of 7%,7% 11%, and 33%Respectively [2,3]. Also, a study conducted in Dar es Salaam, Tanzania, reported a prevalence of 11.5% and an in-hospital mortality rate range of 50% to 70% [2,4–7].

Cryptococcal meningitis (CM) can be treated with various antifungal agents, including fluconazole, voriconazole, amphotericin B, flucytosine, and Posaconazole. However, susceptibility to these agents varies between *Cryptococcus gattii* and *Cryptococcus neoformans*, as well as across different global regions. Historically, CM treatment relied on fluconazole monotherapy at 800–1,200 mg daily doses. Over time, this approach has been associated with the emergence of antifungal resistance, limited fungicidal activity, and high mortality rates [8–11]. In response, the World Health Organisation (WHO) updated guidance to recommend amphotericin B–based induction, specifically single-dose liposomal amphotericin B (10 mg/kg) plus 14 days of flucytosine and fluconazole, which has demonstrated non-inferior survival, improved renal safety, and acceptable tolerability when appropriately monitored [12–14].

Amphotericin B-based regimen is reported to have more rapid clearance of *Cryptococcus* from both serum and cerebrospinal fluid (CSF) cultures. This regimen is highly fungicidal and significantly reduces in-hospital mortality compared to fluconazole monotherapy [10,11,14–16]. Guidelines recommend a single high dose (10 mg/kg) of liposomal amphotericin B, administered in combination with 14 days of flucytosine and fluconazole. This approach has demonstrated non-inferior survival outcomes, improved renal safety, and acceptable tolerability when patients are closely monitored [13].

Tanzania is a WHO member state with a significant burden of HIV [17,18]. Similar to many other SSA countries, the supply of antiretroviral and opportunistic infection

(OI) medications in Tanzania is managed through a vertical program led by the government in collaboration with development partners [19,20]. PHIV access these medications free of charge through designated Care and Treatment Centres (CTCs), which are available in nearly all public healthcare facilities and a few private ones [21]. In alignment with WHO recommendations, Tanzania has updated its national HIV treatment guidelines to include the use of an amphotericin B–based regimen for the treatment of CM among PHIV. However, evidence on the implementation and clinical outcomes of this triple therapy regimen within routine care settings remains limited. Studies have reported several challenges affecting its uptake, including inconsistent supply of amphotericin B and flucytosine, limited healthcare provider skills in amphotericin B administration, concerns about drug toxicity, and inadequate laboratory infrastructure to support safe monitoring [22–24].

A recent study conducted in Tanzania reported that only 22.8% of 45 patients with CM received triple therapy [4]. However, the small sample size limited its ability to compare survival outcomes between patients treated with amphotericin B–based regimens and those receiving fluconazole monotherapy. To address this evidence gap, the present study examined, treatment modalities, outcomes, and factors associated with in-hospital survival among PHIV diagnosed with CM in two Tanzanian regional referral hospitals.

## II. Methods

### A. Study design and setting

This was a hospital-based, cross-sectional study involving retrospective data abstraction. Data were collected from patient records spanning July 2019 to June 2024. The abstraction process was conducted between March and April 2025 at Dodoma Regional Referral Hospital (DRRH) and Singida Regional Referral Hospital (SRRH) in Tanzania. Both facilities are secondary-level referral centres for the central zone, each hosting a CTC for PHIV and internal medicine wards providing inpatient care [25].

### B. Study population

The source population comprised all PHIV admitted to the medical wards of DRRH and SRRH within the study period. From this frame, we identified the CM cohort as those with a clinician diagnosis of CM documented in the chart and/or supporting laboratory evidence (e.g., positive CSF or serum cryptococcal antigen, India ink, or culture), recorded during the index admission.

### C. Sample size

The study included all available medical records of PHIV diagnosed with CM and admitted to the two selected regional referral hospitals from July 2019 to June 2024.

### D. Data collection procedure

Data were collected from patient files and electronic hospital management systems using a structured abstraction tool adapted from a previous study conducted in Dar es Salaam, Tanzania [4]. The tool was used to extract relevant information from patient medical records, including age, sex, marital status, health insurance status, occupation, antiretroviral therapy (ART) status, CD4 count, treatment modality, and treatment outcomes.

The data abstraction process was led by the principal investigator, a second-year Master's student in the Hospital and Clinical Pharmacy program, in collaboration with two trained research assistants. Each assistant held a Bachelor of Pharmacy degree and was stationed at one of the selected RRHs. Before data collection, the research assistants were trained on the study objectives and received a comprehensive orientation on the data abstraction tool.

To ensure consistency and data quality, the principal investigator supervised the initial days of abstraction at each site and conducted ongoing consistency checks and logic/range checks to resolve discrepancies.

## E. Data analysis

Data was initially entered into Microsoft Excel and subsequently imported into IBM SPSS Statistics version 27 for cleaning and analysis. Categorical variables were summarised using frequencies and percentages. Bivariate associations with the primary outcome were screened using Pearson's chi-square; variables with $p \leq 0.20$ entered multivariable models alongside a priori confounders (site and calendar period). For the primary effect estimate, we fitted modified Poisson regression with robust variance to obtain adjusted risk ratios (aRRs) and 95% confidence intervals for being discharged alive comparing amphotericin B–based therapy versus fluconazole monotherapy. We assessed multicollinearity and retained variables with clinical relevance and/or statistical support. Mediator candidates (e.g., completion of induction) were not included in the primary model.

Twenty [20] CM patients who received treatment were excluded from this analysis, as their treatment outcomes could not be determined due to missing or incomplete records (i.e., loss to follow-up). Therefore, the regression analyses were conducted on the remaining patients with clearly documented outcomes, ensuring the validity of the associations identified.

## F. Ethical considerations

Ethical approval was obtained from the MUHAS Research and Ethics Committee (Ref. DA.282/298/01.C/2549). Hospital administrations at DRRH and SRRH granted permission for data access. No personal identifiers (e.g., names, hospital numbers) were abstracted. All procedures complied with institutional policies and the Declaration of Helsinki.

## III. Results

### A. Socio-demographic and clinical characteristics

A total of 159 patient medical records of PHIV with CM were reviewed. The majority of patients, 89 (56.0%), were aged between 36 and 55 years, and females accounted for 89 (56.0%). Most patients, 126 (79.4%), didn't have active health insurance, and 116 (73.0%) were known to be HIV-positive before admission. Additionally, 138 (86.8%) were classified as WHO clinical stage IV. A history of ART default was recorded in 113 (71.1%). CD4 < 200 cells/mm3 was documented in 106 (66.7%) of the patients, and the majority 114 (71.7%) had at least one comorbidity, Table 1.

### B. Cryptococcal meningitis treatment modalities and outcomes

Of the 159 medical records of CM patients reviewed, 71 (44.7%) received an amphotericin B–based regimen. Baseline renal function tests were documented for 52 (73.2%) of these patients, and completion of the induction phase was reported for 48 (67.6%). In-hospital mortality was observed in 74 (46.5%) patients, Table 2.

### C. Factors associated with being discharged alive among study participants

In bivariable analyses, female sex and married/separated status were associated with higher crude likelihood of discharge alive; however, these associations were not statistically significant after adjustment. In the multivariable modified Poisson model, receipt of an amphotericin B–based regimen remained strongly associated with discharge alive compared with fluconazole monotherapy (aRR 4.19, 95% CI 2.46–7.16; p < 0.001), Table 3.

**Table 1. Characteristics of participants in the study (N = 159).**

| Variable | Frequency | Percentage |
|---|---|---|
| **Year of Treatment** | | |
| 2019–2020 | 28 | 17.6% |
| 2021–2022 | 44 | 27.7% |
| 2023–2024 | 87 | 54.7% |
| **Sex** | | |
| Female | 89 | 56.0% |
| Male | 70 | 44.0% |
| **Age in years (Mean: 43.94, Std: 12.28)** | | |
| Below 35 | 42 | 26.4% |
| 36–55 | 89 | 56.0% |
| Above 55 | 28 | 17.6% |
| **Insurance Status** | | |
| Insured | 33 | 20.6% |
| Not Ensured | 126 | 79.4% |
| **Marital Status** | | |
| Divorced | 19 | 12.0% |
| Married | 94 | 59.5% |
| Separated | 8 | 5.1% |
| Single | 23 | 14.6% |
| Widowed | 14 | 8.9% |
| **Occupation** | | |
| Employed | 37 | 23.3% |
| Self Employed | 77 | 48.4% |
| Student | 8 | 5.0% |
| Unemployed | 37 | 23.3% |
| **CD4 Count [Median: 182.81 (59.75–742.00)]** | | |
| ≥ 200 cells/mm3 | 53 | 33.3% |
| < 200 cells/mm3 | 106 | 66.7% |
| **HIV Diagnosis** | | |
| Known HIV Patient at CM treatment | 116 | 73.0% |
| Newly Diagnosed at CM treatment | 43 | 27.0% |
| **WHO Clinical Stage** | | |
| Stage I | 4 | 2.5% |
| Stage II | 8 | 5.0% |
| Stage III | 9 | 5.7% |
| Stage IV | 138 | 86.8% |
| **ART Defaulter** | | |
| Yes | 113 | 71.1% |
| No | 46 | 28.9% |
| **Comorbidity** | | |
| Yes | 45 | 28.3% |
| No | 114 | 71.7% |
| **Comorbidities (N = 45)*** | | |
| Chronic Kidney Disease | 3 | 6.7% |
| Cardiovascular System disorder (Hypertension, Heart failure) | 13 | 28.9% |

*(Continued)*

**Table 1.** (Continued)

| Variable | Frequency | Percentage |
|---|---|---|
| Diabetes Mellitus Type 2 | 2 | 4.4% |
| Malignancy | 2 | 4.4% |
| Opportunistic Infections (Toxoplasmosis, Pneumocystis Jirovecii pneumonia, pulmonary tuberculosis, Herpes) | 25 | 55.6% |

\* One patient may have more than one comorbidity.

**Table 2.** Cryptococcal meningitis treatment modalities, and outcome.

| Treatment Modality (N = 159) | | |
|---|---|---|
| Fluconazole Monotherapy | 88 | 55.30% |
| Amphotericin B- based regimen | 71 | 44.70% |
| **Completion of Induction Phase (N = 71)** | | |
| Yes | 48 | 67.60% |
| No | 23 | 32.40% |
| **Renal Function Tests Done (N = 71)** | | |
| Yes | 52 | 73.20% |
| No | 19 | 26.80% |
| **Treatment Outcomes (N = 139)** | | |
| Discharged alive | 65 | 40.9% |
| In-Hospital Mortality | 74 | 46.5% |
| Lost to follow-up | 19 | 11.9% |
| Referred | 1 | 0.7% |

## IV. Discussion

CM remains a major opportunistic infection among PHIV, and this study examined treatment modalities, outcomes, and factors associated with in-hospital survival among PHIV diagnosed with CM in Tanzania. A total of 159 medical records of PHIV diagnosed with CM were retrieved from July 2019 to June 2024. More than half of the patients (55.3%) were treated with fluconazole monotherapy. Among those who received an amphotericin B–based regimen, renal function tests were conducted for 73.2%, and 67.6% completed the induction phase. Overall, in-hospital mortality was 46.5%. Patients treated with amphotericin B–based regimens were four times more likely to be discharged alive compared to those who received fluconazole monotherapy.

This study demonstrates that CM continues to affect a considerable proportion of PHIV in regions where the disease is endemic. The magnitude observed in our cohort (159 CM patients) is comparable to that reported in Kenya (111 patients), but markedly higher than estimates from previous studies conducted in Dar es Salaam, Tanzania (48 patients), Ethiopia (16 and 14 patients), Cameroon (75 patients), and Sierra Leone (8 patients) [2,4,26–29]. These variations are likely attributable to differences in study design, population characteristics, and data sources; some were cross-sectional, prospective, or retrospective, while others focused on patients attending HIV clinics, hospital admissions, or data retrieved from medical records over time [2].

Although the scale-up of effective ART and the adoption of "test-and-treat" strategies have substantially reduced HIV-related morbidity, mortality, and transmission, CM remains a significant and life-threatening opportunistic infection. Our findings underscore the urgent need to strengthen early HIV diagnosis, timely ART initiation, and sustained adherence support, as well as to monitor emerging HIV-1 resistance patterns that may influence opportunistic infection risk. This

**Table 3. Robust Poisson regression for factors associated with being discharged alive among participants (N = 139).**

| Variable | In hospital mortality (%) | Discharged alive (%) | p – value | Univariate Analysis | | | Multivariable Analysis | | |
|---|---|---|---|---|---|---|---|---|---|
| | | | | cPR | 95% CI | p – value | aPR | 95% CI | p – value |
| **Sex** | | | | | | | | | |
| Female | 36(44.4) | 45(55.6) | 0.014 | 1.611 | 1.075-2.415 | 0.021 | 1.349 | 0.992-1.834 | 0.056 |
| Male | 38(65.5) | 20(34.5) | | 1 | | | 1 | | |
| **Age Groups** | | | | | | | | | |
| <35 | 21(60) | 14(40) | 0.379 | | | | | | |
| 36-55 | 38(48.1) | 41(51.9) | | | | | | | |
| >55 | 15(60) | 10(40) | | | | | | | |
| **Insurance Status** | | | | | | | | | |
| Insured | 15(51.7) | 14(48.3) | 0.854 | | | | | | |
| Not Insured | 59(53.6) | 51(40.5) | | | | | | | |
| **Marital Status** | | | | | | | | | |
| Divorced | 11(64.7) | 6(35.3) | 0.058 | **1.765** | 0.595-5.235 | 0.306 | 1.191 | 0.517-2.7422 | 0.682 |
| Married | 37(45.1) | 45(54.9) | | 2.744 | 1.118-6.737 | 0.028 | 1.928 | 0.909-4.086 | 0.087 |
| Separated | 2(33.3) | 4(66.7) | | 3.333 | 1.174-9.462 | 0.024 | 2.179 | 0.935-5.077 | 0.071 |
| Widowed | 8(57.1) | 6(42.9) | | 2.143 | 0.739-6.216 | 0.161 | 1.379 | 0.611-3.112 | 0.439 |
| Single | 16(80) | 4(20) | | 1 | | | 1 | | |
| **Occupation** | | | | | | | | | |
| Employed | 17(53.1) | 15(46.9) | 0.354 | | | | | | |
| Student | 6(85.7) | 1(14.3) | | | | | | | |
| Self Employed | 34(50) | 34(50) | | | | | | | |
| Unemployed | 17(53.2) | 15(46.9) | | | | | | | |
| **HIV Diagnosis** | | | | | | | | | |
| Known HIV before CM Treatment | 52(51.5) | 49(48.5) | 0.5 | 1.152 | 0.754-1.760 | 0.512 | | | |
| Newly Diagnosed at CM Treatment | 22(57.9) | 16(46.8) | | 1 | | | | | |
| **WHO Clinical Stage** | | | | | | | | | |
| Early stage (I and II) | 3(37.5) | 5(62.5) | 0.358 | 1.365 | 0.773-2.409 | 0.284 | | | |
| Advanced stage (III and IV) | 71(54.2) | 60(45.8) | | 1 | | | | | |
| **ART Defaulter** | | | | | | | | | |
| Non-Defaulter | 24(60) | 16(40) | 0.31 | 1.237 | 0.806-1.899 | 0.33 | | | |
| Defaulter | 50(50.5) | 49(49.5) | | 1 | | | | | |
| **Comorbidity** | | | | | | | | | |
| Without Comorbidity | 54(55.1) | 44(44.9) | 0.496 | | | | | | |
| With Comorbidity | 20(48.8) | 21(51.2) | | | | | | | |
| **CD4 Count Group** | | | | | | | | | |
| <200 | 53(55.8) | 42(44.2) | 0.002 | 2.046 | 1.223-3.422 | 0.006 | 1.299 | 0.857-1.967 | 0.217 |
| >200 | 12(27.3) | 32(72.7) | | 1 | | | 1 | | |
| **Treatment Modalities** | | | | | | | | | |
| Amphotericin B Triple Therapy | 12(16.4) | 61(83.6) | <.0001 | 4.885 | 2.873-8.306 | <0.001 | 4.192 | 2.456-7.155 | <0.001 |
| Fluconazole Monotherapy | 53(80.3) | 13(19.7) | | 1 | | | 1 | | |

Key: cPR: crude Prevalence Ratio, aPR: adjusted Prevalence Ratio, Ref: Reference category.

is consistent with a multicentre study estimating an annual CM burden of approximately 162,500 cases in Africa, largely attributed to delayed ART initiation [6]. Supporting this, studies from Rwanda and South Africa reported a lower prevalence of CM, which was strongly associated with earlier ART initiation [30–32].

Our study revealed notable variability in CM treatment, with most patients receiving fluconazole monotherapy. This aligns with findings from a previous study in Dar es Salaam, Tanzania, where more than 80% of CM patients were treated with fluconazole monotherapy [4]. Such practice, however, contrasts with both national and WHO guidelines, which recommend amphotericin B–based regimens, preferably a combination of amphotericin B, flucytosine, and fluconazole, or amphotericin B with fluconazole when flucytosine is unavailable [14,33]. Process indicators revealed additional implementation gaps whereby baseline renal function testing was documented in 73.2% of amphotericin recipients, and only 67.6% completed the induction phase. Guideline recommendations emphasize renal monitoring due to the drug's nephrotoxicity, and incomplete induction therapy has been strongly linked to poor outcomes [32]. Importantly, our study demonstrated that patients treated with amphotericin B-based regimens were four times more likely to be discharged alive compared to those receiving fluconazole monotherapy. These findings are consistent with evidence from multiple clinical trials showing superior effectiveness of amphotericin B-based therapy, reflected in shorter hospital stays and lower in-hospital mortality [7,12,34–36].

The continued reliance on fluconazole monotherapy and suboptimal completion of amphotericin B therapy likely stems from several factors, including limited awareness of treatment guidelines, inadequate provider training in amphotericin B preparation and administration, irregular drug supply, and the financial burden on patients and the facility. Addressing these systemic barriers is essential for improving CM management and reducing its persistently high mortality in resource-limited settings.

A key limitation of this study is the reliance on data abstracted from patients' medical records, which was constrained by missing information and the inability to seek clarification from prescribers when needed. We mitigated these risks by using a piloted abstraction tool, training and on-site supervision of abstractors, drawing from both paper charts and electronic systems, and applying logic checks during cleaning. We prespecified covariates, adjusted for hospital site and calendar period, excluded patients with unknown disposition from the primary analysis, and ran sensitivity analyses to test robustness. Residual confounding remains possible, particularly from unmeasured severity markers including mental status, intracranial pressure, CrAg titer and lack of time-to-event data. Furthermore, as a purely quantitative study, it does not capture the perspectives and lived experiences of HCPs and patients regarding the burden of CM, treatment practices, outcomes, and the challenges and opportunities that could be leveraged to improve management of this life-threatening opportunistic infection among PHIV.

## V. Conclusion

This study demonstrates that CM remains a leading opportunistic infection causing high mortality among PHIV, with most patients still managed with fluconazole monotherapy despite national and international guideline recommendations. Patients treated with amphotericin B–based regimens had a markedly higher likelihood of survival at discharge, underscoring the urgent need to align clinical practice with evidence-based guidelines. We therefore recommend that CM in PHIV be managed using amphotericin B–based regimens as directed by local and WHO protocols. Additionally, future qualitative research should explore the barriers faced by HCPs, patients, and policymakers in implementing guideline-recommended CM treatment, to inform strategies that can strengthen CM management and reduce associated mortality.

## Supporting information

**S1 File. Analysis outputs generated using the Statistical Package for the Social Sciences (SPSS) software.**
(DOCX)

## Acknowledgments

The authors would like to thank the Dodoma and Singida Regional Referral Hospital administrations for issuing permission for data collection, the staff and research assistants for their support during data abstraction. We further acknowledge all patients whose redacted data made this study possible.

## Author contributions

**Conceptualization:** Mlela Msongela, George Musiba, Manase Kilonzi, Ritah F. Mutagonda.

**Data curation:** Mlela Msongela, Manase Kilonzi.

**Formal analysis:** Mlela Msongela, Juma A. Mohamed.

**Funding acquisition:** Mlela Msongela, Alphonce I. Marealle.

**Investigation:** Mlela Msongela, George Musiba.

**Methodology:** Mlela Msongela, George Musiba, Manase Kilonzi.

**Project administration:** Alphonce I. Marealle, Manase Kilonzi, Ritah F. Mutagonda.

**Resources:** Ritah F. Mutagonda.

**Software:** Juma A. Mohamed.

**Supervision:** George Musiba, Alphonce I. Marealle, Manase Kilonzi.

**Validation:** Juma A. Mohamed, Alphonce I. Marealle, Ritah F. Mutagonda.

**Visualization:** George Musiba, Juma A Mohamed, Ritah F. Mutagonda.

**Writing – original draft:** Mlela Msongela, Manase Kilonzi.

**Writing – review & editing:** Mlela Msongela, George Musiba, Juma A. Mohamed, Alphonce I. Marealle, Manase Kilonzi, Ritah F. Mutagonda.

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
