## [Decision Letter · Decision Letter 0]

29 Sep 2025

Dear Dr. Kilonzi,

Thank you for submitting your manuscript to PLOS ONE. After careful consideration, we feel that it has merit but does not fully meet PLOS ONE’s publication criteria as it currently stands. Therefore, we invite you to submit a revised version of the manuscript that addresses the points raised during the review process.

We look forward to receiving your revised manuscript.

Kind regards,

Felix Bongomin, MB ChB, MSc, MMed, FECMM

Academic Editor

PLOS ONE

Journal Requirements:

3. We note that your Data Availability Statement is currently as follows: All relevant data are within the manuscript and in Supporting Information files.

Reviewers' comments:

Reviewer's Responses to Questions

**Comments to the Author**

1. Is the manuscript technically sound, and do the data support the conclusions?

Reviewer #1: Partly

2. Has the statistical analysis been performed appropriately and rigorously?

Reviewer #1: Yes

3. Have the authors made all data underlying the findings in their manuscript fully available?

Reviewer #1: Yes

4. Is the manuscript presented in an intelligible fashion and written in standard English?

Reviewer #1: Yes

Reviewer #1: In the background, the authors should give reasons as to why, despite the WHO Guidelines, approx half of the patients were receiving fluconazole mono therapy? This could be part of a selection bias.

Equally explain bout the cost of care including the drugs and monitoring, are these provided for free by the government or are bought by the patients. This could equally affect the outcomes.

In the abstract, and manuscript, please remove the emphasis on "prevalence" as this is un-necessary, the title is clear on the objectives (In-hospital mortality and comparison of the treatment regimens).

In the methods and results, please restrict the study to the true study population, ie only those with a diagnosis of CCM.

**Do you want your identity to be public for this peer review?** For information about this choice, including consent withdrawal, please see our Privacy Policy

Reviewer #1: No

---

## [Author Response · Author response to Decision Letter 1]

13 Oct 2025

A response to the reviewers document has been uploaded

---

## [Editor Report · Decision Letter 1]

5 Nov 2025

In-Hospital Survival of Adults with HIV-Associated Cryptococcal Meningitis in Tanzania: A Retrospective Comparison of Amphotericin B-based Regimen and Fluconazole Monotherapy

PONE-D-25-48162R1

Dear Dr. Kilonzi,

We’re pleased to inform you that your manuscript has been judged scientifically suitable for publication and will be formally accepted for publication once it meets all outstanding technical requirements.

Kind regards,

Felix Bongomin, MB ChB, MSc, MMed, FECMM

Academic Editor

PLOS ONE
---

## [Editor Report · Acceptance letter]

PONE-D-25-48162R1

PLOS ONE

Dear Dr. Kilonzi,

I'm pleased to inform you that your manuscript has been deemed suitable for publication in PLOS ONE. Congratulations! Your manuscript is now being handed over to our production team.

Kind regards,

on behalf of

Dr. Felix Bongomin

Academic Editor

PLOS ONE